# Environmentally Sustainable Offset Prints Exposed to Thermal Aging and NO$_2$

Ivana Bolanča Mirković [1,*], Goran Medek [2], Zdenka Bolanča [3] and Milena Reháková [4]

1    Faculty of Graphic Arts, University of Zagreb, 10 000 Zagreb, Croatia
2    Lana Karlovačka Tiskara d.d., 47 000 Karlovac, Croatia; goran.medek@lana.hr
3    Croatian Academy of Engineering, 10 000 Zagreb, Croatia; zbolanca@htz.hr
4    Faculty of Chemical and Food Technology, Slovak University of Technology in Bratislava,
     812 37 Bratislava, Slovakia; milena.rehakova@stuba.sk
*    Correspondence: ivana.bolanca.mirkovic@grf.unizg.hr

**Abstract:** The research aims to find out the crucial factors in the design phase of packaging products, which are related to the determination of environmental influences on sustainable materials. The paper presents the results of research into the influence of environmentally friendly cardboard and the separation of yellow offset ink on the optical properties of prints exposed to thermal aging without and with exposure to NO$_2$. The samples were obtained under real conditions on a Roland 705 printing machine. The colorimetric characteristics of the print and its stability were determined in the research. The research is significant for graphic reproduction in the domain of testing the quality of the print itself, which is defined by certain raster characteristics. The research covers prints in 100%, 70% RTV, 50% RTV, and 30% RTV. The intensity of the tonal experience will depend on the interaction of the substrate with the raster and different types of inks in offset printing as a function of the experimental conditions. The ink characteristics of prints $\Delta L^*$, $\Delta a^*$, $\Delta b^*$, and $\Delta E$ were determined. The research results show that ink I1, with about 80% renewable raw materials, achieves the best stability under the specified experimental conditions.

**Keywords:** sustainability; printing materials; thermal aging; NO$_2$; colorimetry; RTV

## 1. Introduction

The circular economy in the industrial system contributes to the maximum reduction in resource input and waste, emissions, and energy consumption by slowing down, closing, and narrowing material and energy loops [1–3]. Such an economy has its source in a cyclical system in nature, which developed over millions of years [4]. Such rustling is highly specific; there is no waste; and everything that is discarded in one stage is used in the next stage of the cycle. The circular economy is a sustainable alternative to the traditional linear economy, which follows a "take, make, dispose" model [5–7]. The slowdown in the circular economy emphasizes extending the life of products and/or materials, promoting durability, repairability, and renewability [8–11]. By slowing the rate at which products become obsolete or discarded, the overall demand for new resources is reduced. Closing the loops involves designing products and systems to facilitate the recovery and recycling of materials [12–14]. The goal is to reduce the need for new raw materials and maximize the introduction of recycled materials into the production process [15,16]. Narrowing the loops involves optimizing the use of resources within the production process [17].

Many authors deal with terms related to the circular economy and sustainability in their research. Geissdoerfer et al. describe the differences in terms of the concepts of circular economy and sustainability, which contribute not only to the efficiency of application but also to certain limitations [18]. Morales et al. investigated the systematic structure of the interaction between sustainability and the circular economy based on literature data in the period 2004–2021 [19]. Circular economy and sustainability are interrelated but not

interchangeable. Thus, the circular economy aims to make the production process more efficient, reducing and recycling the materials of the production process as much as possible, while sustainability tries to harmonize and manage production resources through greater use of more sustainable or renewable materials [1,20,21]. In order not to break the circular processes in nature, it must be emphasized that the use of sustainable materials must not exceed the amount that nature can replace [22,23].

Sustainability in the field of the graphic industry is mainly manifested in the consumption of paper made from sustainable or new renewable types of raw materials, the use of dyes without volatile organic compounds (VOCs) and heavy metals, the reduction in energy consumption, recycling, as well as in solving problems within the reproduction process itself [24,25]. The commercial advantages of the circular economy approach and the advantages of the circular regenerative approach are in many cases already being used in the printing industry. A special benefit can be achieved through design and in the early stages of product creation [26]. Here, we should mention the design of the product on a modular basis, as well as the use of secondary raw materials, and the design to reduce the consumption of energy, materials, and emissions [27,28]. The circular and sustainable graphic design prioritizes the impact that design can have on the environment during the life cycle of the product, including the closed circular loop of energy and materials [29].

The market for packaging products is constantly growing. The attractiveness and selection of the packaging product are greatly influenced by the design, the choice of materials, and the printing technique used [30,31]. When designing a cardboard packaging product, one must think about the environmental conditions that affect the product. They can have a significant impact on the durability of the print on cardboard packaging products [32,33]. Depending on the type of ink and cardboard and the conditions of storage or transport, the prints on the packaging may be subject to various changes [27]. High levels of humidity can cause smearing of the ink and changes in the structure of the paper or cardboard [34–36]. On the other hand, low humidity can cause the paper to crack, which can negatively affect the legibility of prints. Extreme temperatures can affect the chemical properties of the ink, which can lead to color fading or changes in the texture of prints [37]. UV radiation can cause fading of colors on prints [38–40]. To reduce the environmental conditions of the packaging product, it is important to correctly choose the type of paper or cardboard and the ink that will be used in the production of the product [41,42]. It should be noted that special attention should be paid to packaging for food or pharmaceutical products when choosing materials, due to strict regulations related to the quality of prints.

In addition to the above, the presence of certain gases in the air affects the quality and durability of the print. Burge et al. investigated some types of printing materials and determined their resistance to fading or yellowing caused by nitrogen dioxide at a concentration of 5 ppm for four weeks. Substrates applicable in traditional photography, digital printing, and offset printing faded the least, and black inks are resistant to fading due to exposure to $NO_2$ [43]. The influence of pollutants $SO_2$, $NO_2$, and $O_3$ of different concentrations on the change in properties of seven different papers containing different types of pulp and different acidity was also investigated. The results showed that $NO_2$ affects the decomposition of samples more than $SO_2$ and $O_3$ [44]. Studying the reaction mechanism of $NO_2$ with paper or polymers with a C chain, at elevated temperatures (298 K), it was determined that carbonyl and hydroxyl groups are formed along with nitration products [45]. Slugić et al. investigated the influence of $NO_2$ on the stability of white-colored Inkjet prints [46]. Exposure of $NO_2$ prints was performed in the presence of an accelerated thermal aging process. They proved that multiple layers of white ink caused more fading when exposed for 1 day at 100% HV. After that, further exposure leads to stabilization. Increasing the number of layers of white color causes the creation of low halftone values [46].

In this scientific research, the goal is to determine the colorimetric characteristics of prints and their stability under conditions of dynamic exposure to thermal aging without

and with known concentrations of NO$_2$ as a function of the chemical composition of the printing substrate and inks, including environmental acceptability.

Permanence and durability have been unequivocally established to describe the stability of prints. Permanence primarily refers to the chemical resistance of the impression component and the influence of external factors. Durability depends on the characteristics of raw materials and the material production process as well as environmental conditions. These studies include the influence of the type, proportion, and characteristics of cellulose fibers, fillers, sizing agents, and additives for cardboard as a printing substrate. For the colors used, the research primarily refers to the type and proportion of solvents and the characterization of the yellow pigment. All the above is in accordance with the requirements of offset printing, as a technique used to prepare prints, which are used as samples.

Research in the field of colorimetry is based on the CIE LAB color space. This space is three-dimensional and is defined as a rectangular coordinate system with the edges L* (indicates lightness), a* (red–green), and b* (yellow–blue).

For graphic reproduction, the experience of tonality, which is determined by certain characteristics of the raster, is important. Because of this, the research also includes testing prints at 100% RTV, 70% RTV, 50% RTV, and 30% RTV. The paper presents the results of research related to color attributes, but there are also discussions about the experience of color itself. On the print, the tone value is a term expressed in percentages, which is used to indicate the visual experience of the shade in relation to the printing surface and the value of the solid color. When the base value is zero, it means that there is no tone, and the solid tone value is 100% raster tone value (RTV), i.e., full tone. The tone of ink or tonality of the color of ink quality is determined by the wavelength of light rays, which cause the sensation in the eye and color. The color of tone indicates the type of ink or ink itself.

From the point of view of graphics technology, a significant difference in color ΔE is observed, which represents the deviation of tristimulus information about the color of samples before and after exposure to thermal aging both without and with NO$_2$. The specified color difference is defined by standardized values. The displayed color differences dE* can be described visually. The perceived colorimetric difference can be acceptable to the user of the packaging product.

The application of the results is significant in the design phase of graphics, especially packaging products, and in the formulation of new materials. It can serve as a contribution to the sustainability of resource flows and distribution in modern systems of production and consumption The color fastness of the used pigment is essential for the stability of printing inks, especially at certain stages of the life cycle. The displayed color differences ΔE* can be described visually. The perceived colorimetric difference can be acceptable to the user of the packaging product.

## 2. Materials and Methods

### 2.1. Materials

The prints were made on Kromopak cardboard Količevo, Slovenia (marked S). This substrate belongs to group GC2 according to the cardboard classification according to DIN 19303:2011 [47]. CG2 is a cellulose cardboard made from a layer of mechanically and chemically bleached cellulose. It has extremely high gloss in the GC2 board group, high smoothness < 1 pps, and high degree of brightness (47). The environmentally friendly use of high-quality recycled paper (post-industrial fibers, wood-free white paper, and non-printed paper) together with certified wood pulp is a significant contribution to achieving sustainability goals.

The prints were made with offset ink produced by SunChemical® Europe; they are marked with I1, I2, and I3 marks and have different compositions. The inks used are available as a set of colors offset inks with four processes. Inks marked I1 contain 80% renewable raw materials. Inks marked I2 are plant-based, without mineral oils, dry by penetration, and with a high degree of oxidation. Inks rated I3 are based on conventional

mineral oil and do not contain cobalt-based drying catalysts. These inks are dried by penetration and to a high degree by oxidation. All colors used have C2C Certified Material Health Certificate™, which is a trademark of Cradle-to-Cradle products. These inks are based on an innovative resin/oil combination that dries by absorption and oxidation.

### 2.2. Methods

#### 2.2.1. Design a Test Form

A test print form of 350 × 500 mm format was created (Figure 1). It contained printing elements for determining the quality of color reproduction: standard CMYK RGB stepped wedges, fine raster in low and high tonal values, ISO photo, and a reference cardboard box for the pharmaceutical industry. The largest area is occupied by a standard wedge with 378 fields for creating ICC profiles and 3D gamut.

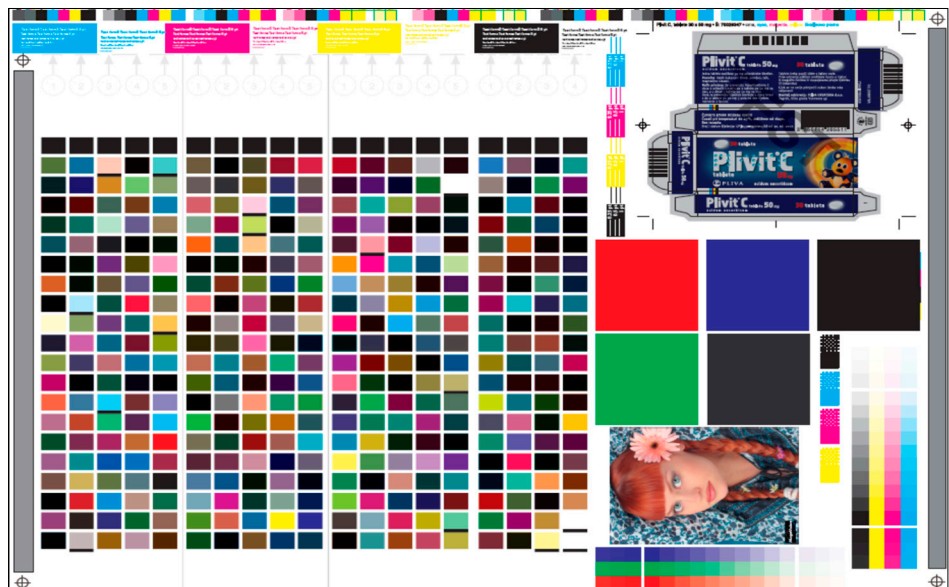

**Figure 1.** Test form.

#### 2.2.2. Offset Printing

The samples were printed on a Roland 705 five-color printing machine that includes a varnishing unit. Even the most demanding cardboard packaging is printed on the machine, and the print is of high quality, and different effects are possible with special colors and varnishes.

The printing form has printing elements and free surfaces almost in the same plane. Printing is possible due to the difference in the physical and chemical properties of printing and free surfaces. Free surfaces are hydrophilic and oleophobic, and printing elements are oleophilic and hydrophobic [48,49]. An aluminum plate coated with a thin photosensitive diazo layer was used. Illumination with a defined light source and development creates a printing form with free surfaces—aluminum oxide and printing elements—copy layer. The printing form is placed on the base cylinder and is in direct contact with the wetting and coloring device. The wetting solution is applied to the entire surface of the printing form and is only accepted on free surfaces [50,51]. Then the ink is applied to the printing elements and transferred to the printing surface via the offset cylinder. Thus, the print is created.

#### 2.2.3. The Method of Accelerated Thermal Aging without and with $NO_2$ Exposure

The method of accelerated thermal aging with and without $NO_2$ according to ISO 5630-5 and ISO 5630-6 was used [52,53]. With this procedure, we determine the influence

of elevated temperature, i.e., elevated temperature and $NO_2$, on the properties of paper and prints to predict stability to long-term natural aging.

Samples for analysis were placed in Pyrex R bottles and conditioned for 24 h at 230 °C and 50% RH. They were hermetically sealed with polyphenylsiloxane using a Viton R seal. $NO_2$ was introduced into certain bottles in a concentration of 100 ppm or 800 ppm. The samples were aged for 1, 2, 3, 4, and 5 days.

### 2.2.4. Determination of Optical Properties

The yellow separation of prints (SI1, SI2, SI3) before and after exposure to elevated temperature, elevated temperature with 100 ppm $NO_2$, and elevated temperature with 800 ppm $NO_2$ for periods of 1, 2, 3, 4, and 5 days were examined for their optical properties (Table 1). Yellow separation was measured on 100%, 70%, 50%, and 30% raster tone values (RTVs). They were analyzed using X-Rite DTP and Technidyne Color Tauch Spectrophotometer. The X-Rite DTP spectrophotometer by connecting to the ColorShop 2.6 application enables the measurement of RTV, Lab, XYZ, xyY, LCh, Luv, RGB, color density of different status, spectral reflection, and spectral transmission. This paper presents only part of the results obtained with this device, in accordance with ISO/CIE 11664-1:2019 and ISO/CIE 11664-6 2022 [54,55].

**Table 1.** Labels of the samples.

| Type of Sample | Ink | Label |
|---|---|---|
| Prints | I1 | SI1 |
| | I2 | SI2 |
| | I3 | SI3 |

The Technidyne Color Touch spectrophotometer quickly and accurately measures the optical characteristics of pulp and paper: brightness, opacity, and whiteness [56–58].

### 3. Results

The results of the research on the colorimetric characteristics of the prints, obtained under real printing conditions in combinations of GC2 cardboard/ink with different chemical compositions, are presented. Raster tone values are 100%, 70%, 50%, and 30%. The dynamics of exposure of prints is 1, 2, 3, 4, and 5 days of thermal aging without and with the presence of different concentrations of $NO_2$.

Each color can be described with CIE L*a*b* value. An important variable in the expression for calculating the total color difference in the CIE system is $\Delta L^*$. Chromatic axis L has values from 0 to 100 (black at 0, white at 100). To evaluate the results, it is important to emphasize that a positive value of $\Delta L^*$ means that the exposed print is lighter than the unexposed one.

Figure 2 shows the $\Delta L^*$ values of yellow ink separation for 100%, 70%, 50%, and 30% RTV prints exposed to thermally accelerated aging without and with 100 ppm $NO_2$ and 800 ppm $NO_2$.

The research results show that prints SI2 leads to faster fading of $\Delta L^*$ compared to SI1 and SI3 when exposed to accelerated thermal aging for 1 day ($\Delta L^*_{SI2,\ 100\%\ RTV,\ 1\ day} = -1.65$) (Figure 2). By increasing the exposure time to 5 days, the highest negative value of $\Delta L^*$ is achieved by sample SI3 ($\Delta L^*_{SI3,\ 100\%\ RTV,\ 5\ days} = -3.04$). The biggest difference between 1 day of exposure and 5 days is achieved by prints SI3 with a value of $-\Delta L$ 1.79. Analyzing the entire period of exposure of print to accelerated thermal aging, it was determined that print SI1 achieves less negative $-\Delta L^*$ values, followed by SI3 print and then SI2. The obtained data speak of their stability under experimental conditions. By decreasing the raster tonal values of the prints, aging contributes to a decrease in the negative $\Delta L^*$ values, which is expected considering the decrease in the dotted area covered by the ink.

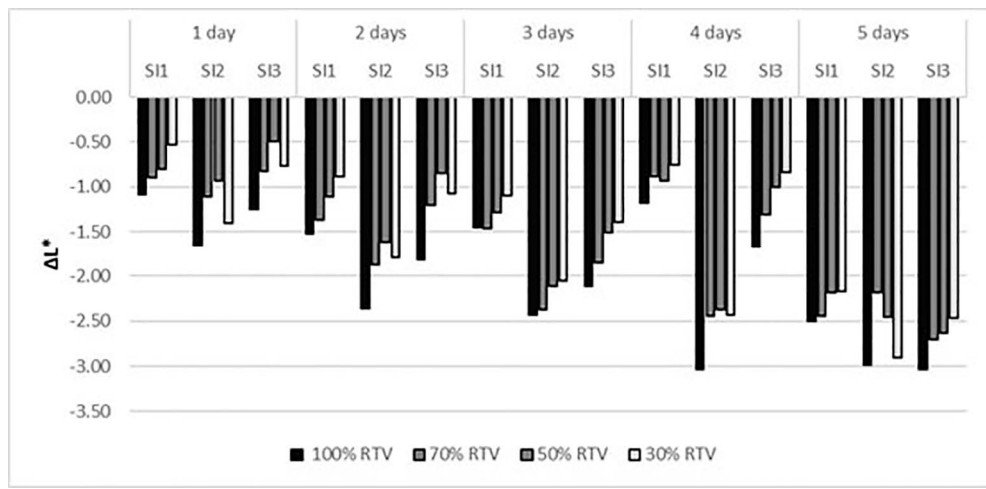

**Figure 2.** Influence of dynamics of accelerated thermal aging on ΔL* values for samples SI1, SI2, and SI3 for 100%, 70%, 50%, and 30% RTV.

The research results show that the negative values of ΔL* decrease in almost all samples compared to the period of exposure to accelerated thermal aging in the presence of 100 ppm $NO_2$ (Figure 3). The drop in the value of ΔL* ranges from 0.02 to 0. The lowest stability according to the exposure in the mentioned experimental conditions was determined for the SI2 sample with a full tone. The difference ΔL* for the print exposed to accelerated thermal aging with 100% and without $NO_2$ after 3 days of exposure is ΔL* = 0.69. Under the experimental conditions stated here, the largest increase in negative ΔL* values occurs on the third and fourth days of sample exposure. The smallest increase in the negative value was found for inks SI3, followed by SI1.

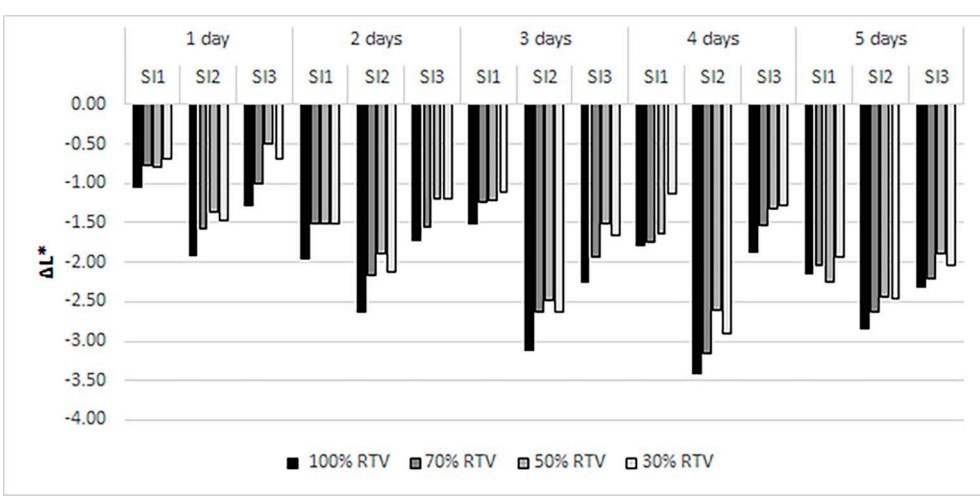

**Figure 3.** Influence of dynamics of accelerated thermal aging and exposition of 100 ppm $NO_2$ on ΔL* values for samples SI1, SI2, and SI3 for 100%, 70%, 50%, and 30% RTV.

By exposing the samples for 1 day to accelerated thermal aging in the presence of 800 ppm $NO_2$ and 100% RTV, negative -ΔL values are determined as follows: SI1 = 1.35, SI2 = 1.89, and SI3 = 1.42 (Figure 4). After 5 days of exposure, the negative ΔL values increase as follows: SI1 = 2.38, SI2 = 2.62, and SI3 = 3.27. The percentage of ΔL values is as follows: SI1 = 55.7%, SI2 = 27.9%, and SI3 = 56.3.%. By reducing the raster tonal value from 100% RTV of the prints exposed to thermal aging with 800 ppm $NO_2$ for 1 day, there is a decrease in ΔL* as follows: 76.3% (70% RTV), 62.9% (50% RTV), and 57.0% (30% RTV).

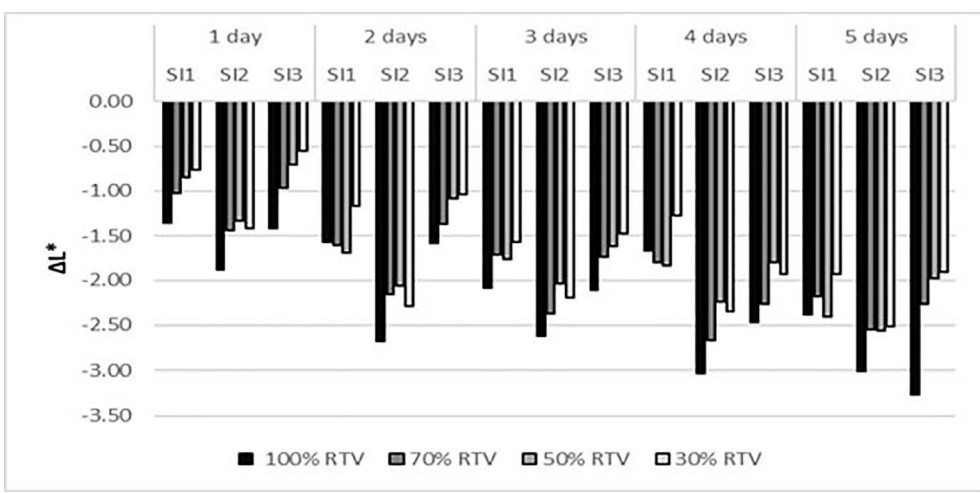

**Figure 4.** Influence of dynamics of accelerated thermal aging and exposition of 800 ppm $NO_2$ on $\Delta L^*$ values for samples SI1, SI2, and SI3 for 100%, 70%, 50%, and 30% RTV.

According to the CIEXYZ color space, the positive chromatic axis a is directed towards the red stimulus and the negative towards the green. The difference in a* value between the sample exposed to thermal aging with or without $NO_2$ and the sample not exposed is presented with $\Delta a^*$ value.

Based on the results of the research, it was found that SI1, SI2, and SI3 samples were found with yellow colors with tone values of 100%, 70%, 50%, and 30% of accelerated thermal aging without the presence of a gaseous $NO_2$ have positive values $\Delta a^*$ (Figure 5). The results indicate a larger or smaller shift towards the red area, which is a function of color composition, raster tone value, and the time of exposure to accelerated heat aging. The results show that samples SI1, SI2, and SI3 exposed to thermal aging by increasing the aging period have a higher $\Delta a^*$ value. The differences in $\Delta a^*$ for samples exposed for 5 days compared to 1 day are as follows: $\Delta a^*$ SI1 = 1.12, $\Delta a^*$ SI2 = 1.06, and $\Delta a^*$ SI3 = 1.54.

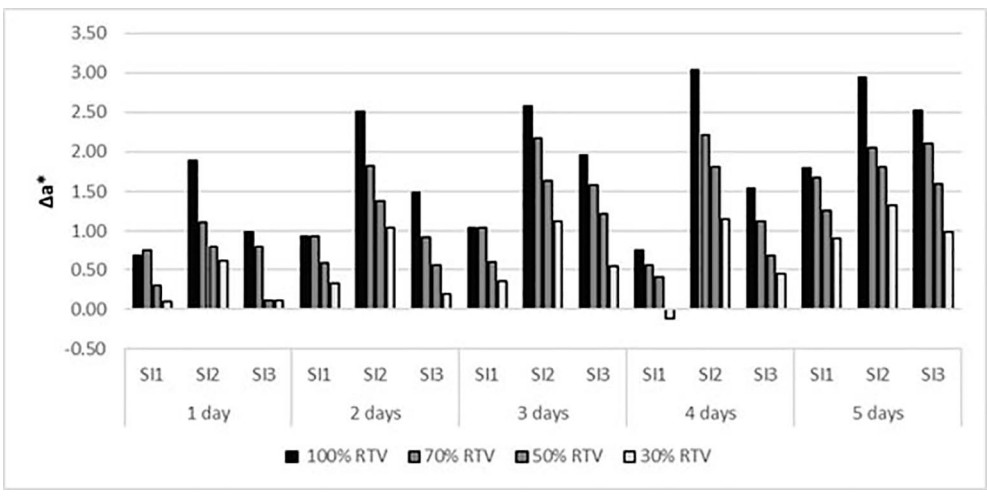

**Figure 5.** Influence of dynamics of accelerated thermal aging on $\Delta a^*$ values for samples SI1, SI2, and SI3 for 100%, 70%, 50%, and 30% RTV.

The results in Figure 6 show that prints SI1, SI2, and SI3 with 100% RTV after exposure to accelerated thermal aging in the presence of 100 pp $NO_2$ have higher $\Delta a^*$. Decreasing the raster tonal value of prints decreases $\Delta a^*$ and increasing with the days of exposure of prints increases $\Delta a^*$ ($\Delta a^*$ SI1, RTV 100% 1 day = 0.59; $\Delta a^*$ SI1, 30% 1 day = 0.13; $\Delta a^*$ SI1, 100%RTV,5 days = 1.31; $\Delta a^*$ SI1, 30%5 days = 0.39).

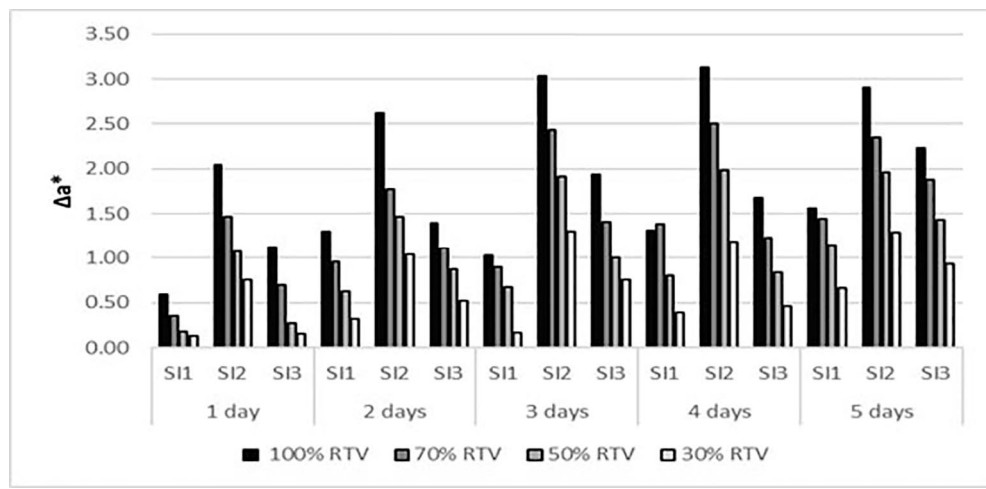

**Figure 6.** Influence of dynamics of accelerated thermal aging and exposition of 100 ppm $NO_2$ on $\Delta a^*$ values for samples SI1, SI2, and SI3 for 100%, 70%, 50%, and 30% RTV.

The effect of increasing the concentration of $NO_2$ from 100 ppm to 800 ppm does not significantly affect the value of $\Delta a^*$ during the exposure of the sample to the process of accelerated thermal aging for a period of 1 day and 100% RTV (Figures 6 and 7). It must be emphasized that the sample exposure procedure under the mentioned conditions has the greatest effect on the samples printed with SI2 ink. When studying the influence of concentration on samples with 100% RTV, in subsequent exposure intervals (from 2 to 5 days), it can be observed that at 800 ppm, SI2 reaches its maximum value $\Delta a^*$ when the sample is exposed for 2 days, and at a concentration of 100 ppm when the sample is exposed for 3 days.

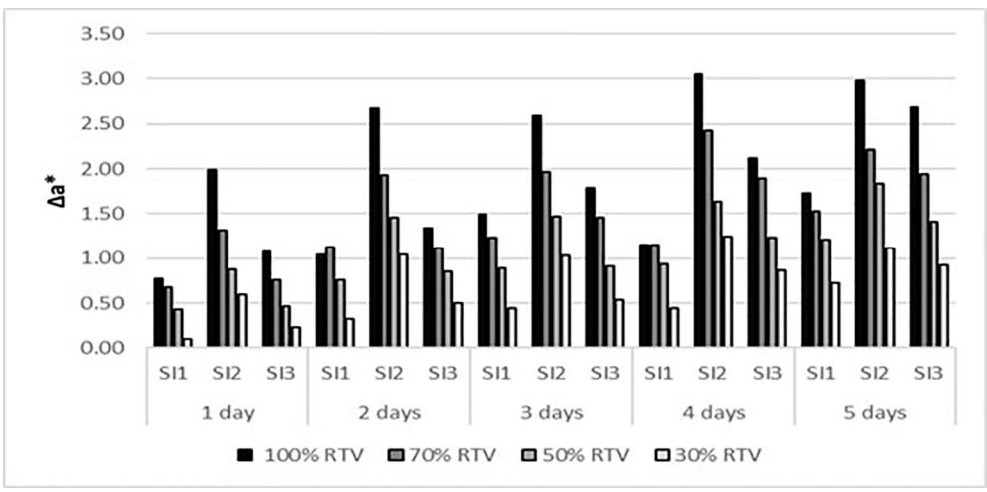

**Figure 7.** Influence of dynamics of accelerated thermal aging and exposition of 800 ppm $NO_2$ on $\Delta a^*$ values for samples SI1, SI2, and SI3 for 100%, 70%, 50%, and 30% RTV.

One of the defined axes in the CIELab color space is the chromatic axis b with a yellow–blue orientation. The positive b-axis is in the direction of the yellow stimulus, and the negative b-axis is in the direction of the blue stimulus. The $\Delta b^*$ value is the difference in the b* value of the exposed print minus the b* value of the unexposed print.

Exposure of prints SI1 and SI2 to accelerated thermal aging without $NO_2$ revealed negative values of $\Delta b^*$ for prints with 100% RTV and 70% RTV, which means orientation towards the yellow tristimulus (Figure 8). Print SI3 with 50% RTV after exposure for 1 day and 5 days has a positive $\Delta b^*$ value, i.e., towards blue tones. Prints SI1, SI2, and SI3 with 30% RTV have a positive $\Delta b^*$, which means an orientation in the direction of the blue

stimulus. From all the above, it can be seen that the RTVs have a great influence on the sign of the Δb* value.

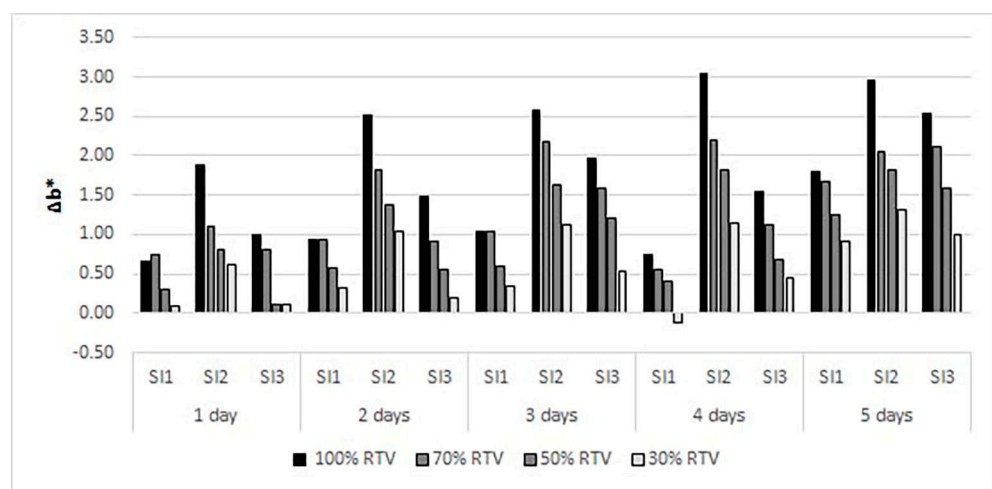

**Figure 8.** Influence of dynamics of accelerated thermal aging on Δb* values for samples SI1, SI2, and SI3 for 100%, 70%, 50%, and 30% RTV.

The largest increase in Δb* between 1-day and 5-day thermal aging occurs in the imprint SI3 (with 100% RTV = 2.42), compared to the other two prints SI1 and SI2. In light tones as 30% RTV, the greatest difference between 1-day and 5-day accelerated thermal aging occurs in the imprint SI2 ($\Delta b^*_{SI2, RTV\ 30\%\ 5\ days} - \Delta b^*_{SI2, RTV\ 30\%\ 1day} = 3.58$).

Exposure of prints SI1, SI2, and SI3 with 100% RTV and 70% RTV to an accelerated thermal aging process with 100 ppm $NO_2$ determines the negative sign of Δb* (Figure 9). When exposing prints SI1, SI2, and SI3 with 100% RTV, it can be noticed that the value of Δb* for samples printed with ink S1 and S3 reaches its maximum negative value on the fifth day of exposure, while with ink S2 it happens already after the second day of exposure. Exposure of prints SI1, SI2, and SI3 of lower raster tone values 50% and 30% to accelerated thermal aging with 100 ppm $NO_2$ results in the value of the difference in the chromatic coefficients b* with a positive sign.

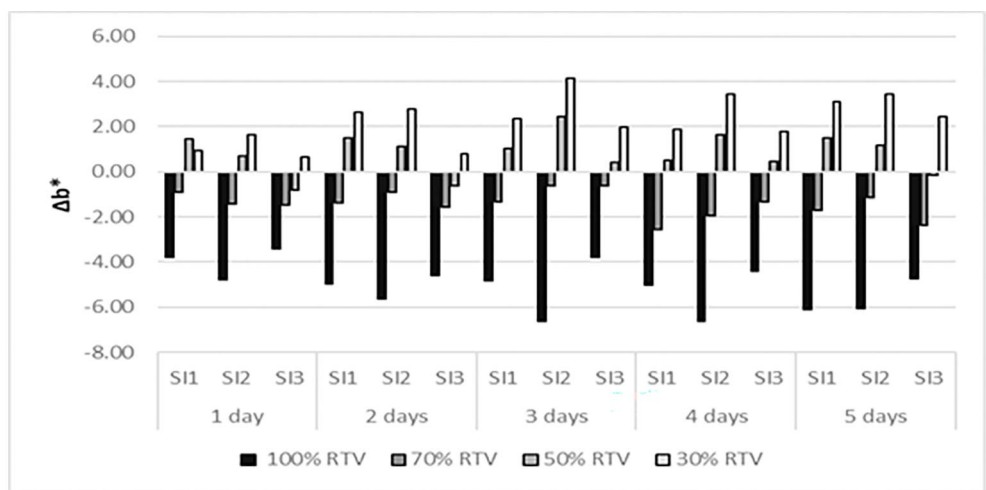

**Figure 9.** Influence of dynamics of accelerated thermal aging and exposition of 100 ppm $NO_2$ on Δb* values for samples SI1, SI2, and SI3 for 100%, 70%, 50%, and 30% RTV.

Based on the results of the research, it was determined that prints SI1, SI2, and SI3 with yellow ink with raster tone values of 100% and 70% and accelerated thermal aging with a concentration of 800 ppm $NO_2$ have negative Δb* values (Figure 10). By

increasing the number of days of exposure of prints, the value of $\Delta b^*$ increases as follows: $\Delta b^*_{\text{SI1, RTV 100\%,1 day}} = -4.28$; $\Delta b^*_{\text{SI2, RTV 100\% 1 day}} = -4.85$; $\Delta b^*_{\text{SI3, RTV 100\% 1 day}} = -3.33$; $\Delta b^*_{\text{SI1, RTV 100\% 5 days}} = -6.21$; $\Delta b^*_{\text{SI2, RTV 100\% 5 days}} = -6.12$; $\Delta b^*_{\text{SI3, RTV 100\% 5 days}} = -5.48$. The largest increase in $\Delta b^*$ occurred between the 1-day and 5-day thermal aging with 800 ppm $NO_2$ on the SI3 print ($\Delta b^*_{\text{SI3, RTV 100\% 5 days}} - \Delta b^*_{\text{SI3, RTV 100\%, 1 day}} = 2.15$). Prints SI1, SI2, and SI3 with 50% and 30% RTV exposed to accelerated thermal aging in the presence of 800 ppm $NO_2$ have positive $\Delta b^*$ values.

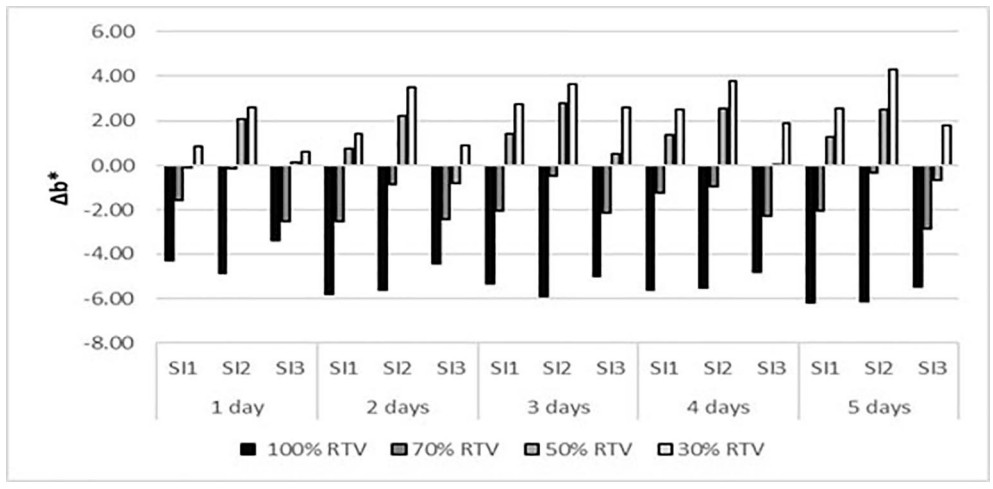

**Figure 10.** Influence of dynamics of accelerated thermal aging and exposition of 800 ppm $NO_2$ on $\Delta b^*$ values for samples SI1, SI2, and SI3 for 100%, 70%, 50%, and 30% RTV.

The $\Delta E$ color difference is important for graphic reproduction because it compares two tones and displays information about the reproduction quality. It refers to the deviation in the reproduction of the original and indirectly shows the deviation from the tristimulus values corresponding to the perception of color in the human eye. It defines the difference between coordinates for two reference positions and compares them, and in this paper, it is used as the difference between unexposed prints and prints exposed to accelerated aging.

The research results show that print with SI1 yellow ink, 100% RTV, and 30% RTV has mostly the lowest $\Delta E$ value compared to SI2 and SI3 for exposure to accelerated thermal aging without the presence of $NO_2$ $\Delta E$ (Figure 11). The same conclusion applies to the 5-day exposure of the above prints.

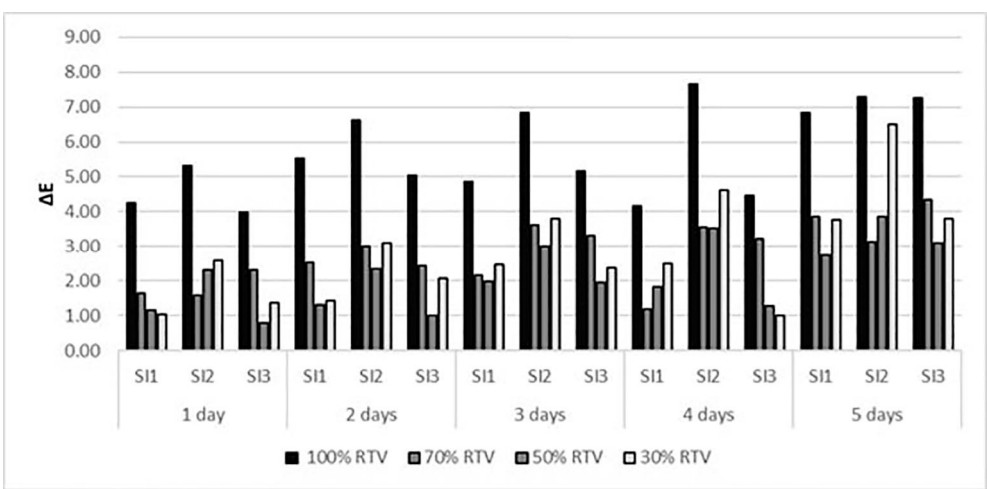

**Figure 11.** Influence of dynamics of accelerated thermal aging on $\Delta E$ values for samples SI1, SI2, and SI3 for 100%, 70%, 50%, and 30% RTV.

However, $\Delta E$ values after exposure of full-tone samples to accelerated thermal aging change as follows: $\Delta E_{SI1, \text{ RTV } 100\% \text{ 5 days}} = 6.82$; $\Delta E_{SI2, \text{ RTV } 100\%, \text{ 5 days}} = 7.30$; $\Delta E_{SI3, \text{ RTV } 100\% \text{ 5 days}} = 7.30$. By reducing the raster tonal value to 70%, 50%, and 30%, $\Delta E$ decreases, but not always in a regular sequence: $\Delta E_{SI1, \text{ RTV } 70\%, \text{ 1 day}} = 1.66$; $\Delta E_{SI1, \text{ RTV } 50\%, \text{ 1 day}} = 1.16$; $\Delta E_{SI1, \text{ RTV } 30\%, \text{ 1 day}} = 1.05$; $\Delta E_{SI2, \text{ RTV } 70\%, \text{ 1 day}} = 1.58$; $\Delta E_{SI2, \text{ RTV } 50\%, \text{ 1 day}} = 2.31$; $\Delta E_{SI2, \text{ RTV } 30\%, \text{ 1 day}} = 2.58$; $\Delta E_{SI3, \text{ RTV } 70\%, \text{ 1 day}} = 2.32$; $\Delta E_{SI3, \text{ RTV } 50\%, \text{ 1 day}} = 0.78$; $\Delta E_{SI3, \text{ RTV } 30\%, \text{ 1 day}} = 1.37$.

In a series of SI1, SI2, and SI3 samples exposed to accelerated thermal aging and a $NO_2$ concentration of 100 ppm, SI3 has the lowest value of $\Delta E$ (Figure 12). The mentioned value of $\Delta E$ for samples with a raster tonal value of 100% fluctuates with exposure time as follows: $\Delta E_{SI3, \text{ 1 day}} = 3.79$; $\Delta E_{SI3 \text{ 2 days}} = 5.08$; $\Delta E_{SI3.3 \text{ days}} = 4.80$; $\Delta E_{SI3.4 \text{ day}} = 5.06$; $\Delta E_{SI3, \text{ 5 days}} = 5.71$. When providing prints with a lower raster tone value (50% and 30% RTV) in all exposure periods, it can be noticed that the $\Delta E$ values are lower than those for higher RTV (100% and 70% RTV).

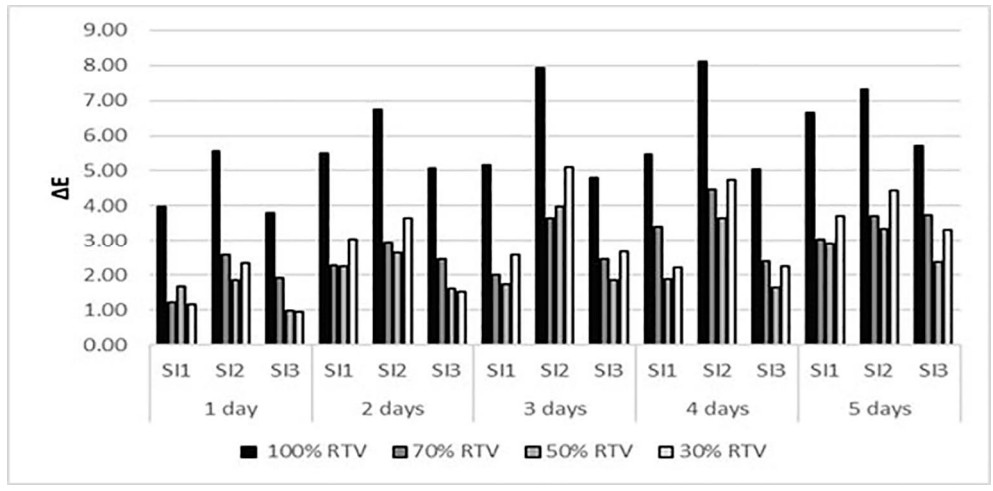

**Figure 12.** Influence of dynamics of accelerated thermal aging and exposition of 100 ppm $NO_2$ on $\Delta E$ values for samples SI1, SI2, and SI3 for 100%, 70%, 50%, and 30% RTV.

The difference in $\Delta E$ of samples SI1, SI2, and SI3 exposed to accelerated thermal aging and 100 ppm $NO_2$ during 1 day of exposure with 100% RTV compared to 70% RTV, 50% RTV, and 30% RTV is shown as follows: $\Delta E_{SI1, \text{ RTV } 100\%} - \Delta E_{SI1, \text{ RTV } 70\%} = 2.75$; $\Delta E_{SI1, \text{ RTV } 100\%} - \Delta E_{SI1, \text{ RTV } 50\%} = 2.51$; $\Delta E_{SI1, \text{ RTV } 100\%} - \Delta E_{SI1, \text{ RTV } 30\%} = 2.31$. The research results show that the greatest influence on the raster tonal value within the described experimental conditions was observed when print SI2 was used.

Figure 13 shows that the smallest $\Delta E$ belongs to the print SI3 exposed to thermal aging with 800 ppm $NO_2$ for 1 day ($\Delta E_{SI3, \text{ RTV } 100\%} = 3.78$), following the $\Delta E$ SI1 value ($\Delta E_{SI1, \text{ RTV } 100\%} = 4.55$) and then sample SI2 ($\Delta E_{SI2, \text{ RTV } 100\%} = 5.52$). The highest value of $\Delta E$ was determined after a 5-day exposure to an imprint of SI2 100% RTV exposed to accelerated thermal aging of 800 ppm $NO_2$ ($\Delta E_{SI2, \text{ RTV } 100\%} = 7.74$). The SI1 100% RTV print exposed to accelerated thermal aging for 5 days with 100 ppm $NO_2$ has a 0.21 lower $\Delta E$ compared to the print aged by the same process but at a concentration of 800 ppm $NO_2$. When comparing the values of $\Delta E$ values for samples exposed to accelerated thermal aging and exposure to $NO_2$ with concentrations of 100 ppm and 800 ppm, it can be noticed that the values for lower staining densities are higher when samples are exposed to concentrations of 800 ppm.

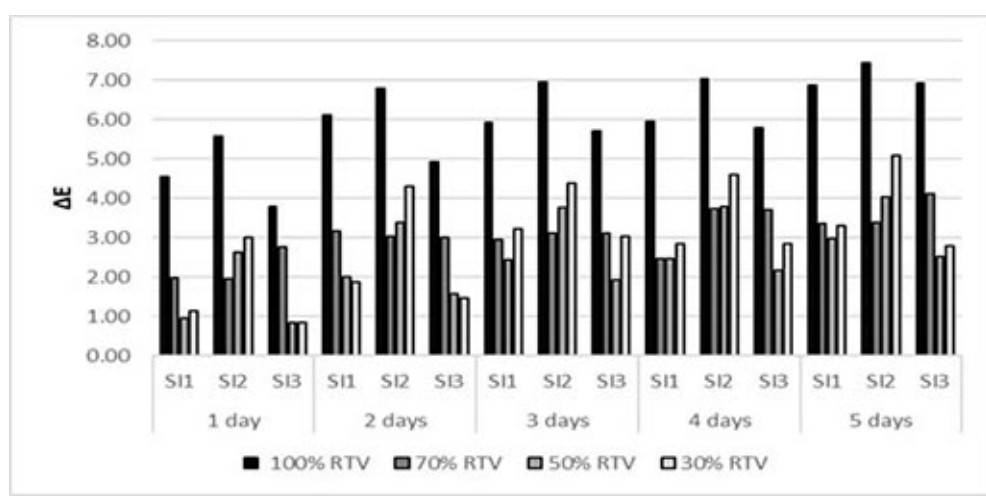

**Figure 13.** Influence of dynamics of accelerated thermal aging and exposition of 800 ppm $NO_2$ on $\Delta E$ values for samples SI1, SI2, and SI3 for 100%, 70%, 50%, and 30% RTV.

Due to the complexity of the work, only for some of the obtained key results, we are showing the limits of the units that can be expected (Tables 2–4).

**Table 2.** Sample SI1, RTV 100%, exposed for 5 days.

|  |  | L* | a* | b* |
|---|---|---|---|---|
| Prints Accelerated thermal aging | Standard deviation | 0.025 | 0.025 | 0.025 |
|  | Variance | 0.015 | 0.001 | 0.001 |
| Accelerated thermal aging + 100 ppm $NO_2$ | Standard deviation | 0.020 | 0.006 | 0.032 |
|  | Variance | 0.001 | 0.001 | 0.001 |
| Accelerated thermal aging + 800 ppm $NO_2$ | Standard deviation | 0.025 | 0.015 | 0.045 |
|  | Variance | 0.001 | 0.001 | 0.001 |

**Table 3.** Sample SI2, RTV 100%, exposed for 5 days.

|  |  | L* | a* | b* |
|---|---|---|---|---|
| Prints Accelerated thermal aging | Standard deviation | 0.020 | 0.006 | 0.026 |
|  | Variance | 0.001 | 0.001 | 0.001 |
| Accelerated thermal aging + 100 ppm $NO_2$ | Standard deviation | 0.025 | 0.010 | 0.040 |
|  | Variance | 0.001 | 0.001 | 0.001 |
| Accelerated thermal aging + 800 ppm $NO_2$ | Standard deviation | 0.025 | 0.006 | 0.036 |
|  | Variance | 0.001 | 0.001 | 0.001 |

**Table 4.** Sample SI3, RTV 100%, exposed for 5 days.

|  |  | L* | a* | b* |
|---|---|---|---|---|
| Prints Accelerated thermal aging | Standard deviation | 0.025 | 0.006 | 0.020 |
|  | Variance | 0.001 | 0.001 | 0.001 |
| Accelerated thermal aging + 100 ppm $NO_2$ | Standard deviation | 0.015 | 0.010 | 0.049 |
|  | Variance | 0.001 | 0.001 | 0.001 |
| Accelerated thermal aging + 800 ppm $NO_2$ | Standard deviation | 0.053 | 0.006 | 0.031 |
|  | Variance | 0.001 | 0.001 | 0.001 |

Variance is a measure of the dispersion of measurement variables, and if the standard deviation is small, the arithmetic mean represents the results well.

## 4. Discussion

Deterioration of aged paper and prints can manifest in the chemical, physical, optical, and mechanical properties. The technique for the study of degradation can be natural or artificial aging. The accelerated aging test is established to assess material stability or durability in a short time frame, estimating potential long-term serviceability of the material under conditions of use and clarifying the chemical reactions involved in the degradation mechanism [59,60]. Porck has published an overview of the possibilities and limitations of artificial aging analysis of paper [61]. Research by a group of authors found that artificial aging in confined spaces better resembles the natural aging of paper [62,63]. Calvini et al. proposed a model that explains the effects on the kinetic equations of different ways of aging from initial oxidation and procedures with and without reduction of oxidized groups. Kinetic equations explain various aging methods including acid hydrolysis, vented oven aging, and closed vessel aging [63].

Samples for our analysis were placed in Pyrex R bottles and were hermetically sealed with polyphenylsiloxane using a Viton. seal.

Erhard and Mecklendburg discuss the kinetics of aging and its application in accelerated aging in the context of accelerated versus natural aging [64]. Based on their results, Lojewski et al. determined that to accelerate the aging process caused by hydrolysis, oxygen-free water vapor should be applied [65]. In this area, it is especially necessary to mention the Zervos study with the presentation of various aspects of natural and accelerated aging of pulp and paper, including the chemistry of aging, the effect of aging on the mechanical, structural, and optical properties of cellulose, the application of accelerated aging methods, as well as the kinetics of aging [66].

The influence of air pollution on the stability of paper and prints was more researched in the last decade of the 20th century in the context of museums and libraries. In the literature, there are many works on the impact of pollutants on printing media and less on prints in the last decade of the 20th century. The most common topics in research are as follows: the synergistic effects of air pollutants and climate on the stability of cellulosic materials; the influence of $SO_2$, $NO_2$, and $O_3$ on the stability of cellulosic materials in the process of accelerated aging; exposure of newsprint to $O_3$, $SO_2$, and $NO_2$ without the presence of light with 60% humidity to ambient levels; accelerated exposure of cellulose-containing materials to air pollutants [67–71].

One of the areas of research is the stability of prints from different printing techniques: light fastness properties of different digital prints, gas and light fading of inkjet prints, and permanence of conventional and digital offset prints [72–74].

As our research relates to offset printing prints in real conditions with the separation of yellow ink, the results will be clarified in relation to the pigment of ink under the conditions of the experiment. Yellow diazo pigments contain a conjugated aromatic structure with different substituents. In the process of photolysis, photon absorption occurs, which breaks the chemical bond. Then the sigma and pi electrons at the anti-binding energy level move to the binding energy level with the photon energy. The bond from which the excited electron comes is destroyed, and the conjugation needed for chromophores is lost [72].

Oxygen is essential in this process when a molecule absorbs a photon. A reaction with oxygen in the ground state is possible, resulting in an excited singlet state, which can react with double bonds, which leads to the results shown.

The effect of optical brighteners and coating substances on the light fastness of inks is known [73].

A highly porous material is very sensitive to gas pollutants such as $NO_2$ absorption and then, the oxidative pathway of pigment begins. Different sensitivity on $NO_2$ will depend on the type of printing substrate, type of ink, and relative humidity.

Vegetable oil plays a significant role in inks for offset printing. These materials are colloidal suspensions of pigments in an oleo-resinous vehicle. The ink vehicle must ensure pigment wetting and transfer, film formulation after printing and drying, and pigment protection during the life cycle. Linseed and several other vegetable oils possess the property of turning progressively from liquid to solid, upon exposure to the air due to an oxidation-polymerization induced by alkenyl moieties. Moreover, pigments can be easily dispersed in these oils, which explains their extensive use.

Increasing ecological sustainability in this segment refers to the substitution of vegetable oils with vegetable esters—products between fatty acids obtained from vegetable oils and alcohol. The similarity in the shape of the molecules ensures a viscosity closer to mineral derivatives than to triglycerides. The consequence of the mentioned is a faster absorption of ink into the paper, which affects the hardening of ink, greater dissolving power of resins, fewer problems with emulsification, and better interaction of the ink with the wetting solution. In ink formulations, vegetable oils may be introduced unaltered or modified, in combination with other components, e.g., resins. They are typically based on mixtures of natural triglycerides bearing long aliphatic chains with varying degrees of unsaturation, lack of conjugation [74–76]. For ink purposes, it is convenient to classify vegetable oils according to their drying ability, which is connected to the number and the relative position (presence or lack of conjugation) of the unsaturation along the fatty acid chain.

The presented discussion further clarifies our results and is consistent with them.

## 5. Conclusions

The circular economy encourages a shift towards a more regenerative and sustainable approach to production through innovation in product design, production processes, and business models [77]. Embracing a circular economy can lead to a more resilient and resource-efficient industrial system, contributing to long-term environmental and economic sustainability. In this paper, the selection of materials and printing techniques plays an important role, which can contribute to the durability of the product, which is exposed to environmental influences daily. The increase in the concentration of nitrogen dioxide ($NO_2$) in the air is most often related to human activities, traffic in vehicles that use fossil fuels, and industrial processes that include the burning of fossil fuels, the production of chemicals, or the processing of metals. Because of all the above, it is extremely important to determine the impact of $NO_2$ on prints, which is the subject of this research.

This paper shows the difference between the unexposed sample (standard sample) and the sample exposed to accelerated thermal aging and accelerated thermal aging with 100 ppm $NO_2$ and 800 ppm $NO_2$, respectively. By processing the obtained measurement results and studying the measurement parameters $\Delta L^*$, $\Delta a^*$, $\Delta b^*$, and $\Delta E$, a trend can be noticed in the results by studying the ink composition I1, I2, and I3. The best results are shown by samples SI1, followed by prints SI3 and SI2.

The research results show that all samples have negative $\Delta L^*$ values. Reducing the RTV print in a sequence of 100%, 70%, 50%, and 30% reduces the $\Delta L^*$ of the print (for accelerated thermal aging SI1, 100% RTV = −1.09; 70% RTV −0.90; 50% RTV 0.80; 30% RTV-- 0.53). The relationship established applies to all prints of this series. When studying the results of the difference in chromatic coefficients $\Delta a^*$, it can be noticed that all samples with 100% RTV, 70% RTV, 50% RTV, and 30% RTV exposed in the previously described conditions have positive $\Delta a^*$ values, which means that they are in the red area. The study of this parameter showed a certain regularity concerning RTV, with the decrease in the RTV of prints decreasing $\Delta a^*$ and an increase in the days of exposure of prints increasing $\Delta a^*$. In addition, it should be emphasized that prints with 30% RTV are located closer to the center of the chromatic axes, which means that they have reduced chromaticity.

Prints with 100% RTV and 70% RTV exposure to accelerated thermal aging, as well as thermal aging with 100 ppm and 800 ppm $NO_2$, have a positive value $\Delta b^*$, which means they are in the yellow color space. Prints with 50% RTV and 30% RTV exposure to

accelerated thermal aging, as well as thermal aging with 100 ppm and 800 ppm $NO_2$, have negative value $\Delta b^*$ values, which means they are in the blue color space.

The highest measured value for $\Delta E$ is 8.10 for the sample exposed to accelerated thermal aging for 4 days with 100 ppm $NO_2$. Each series of prints consisted of 176 samples, and $\Delta E$ in the range 6–7 was measured only for 10 samples, and in the range 7–8, there were 7 samples. Other samples had lower $\Delta E$ values measured. The permissible limits of deviation $\Delta E$ can be determined by the industry with its technology, according to a specific determination. The deviation limits of $\Delta E$ can be determined by the graphic industry in the domain of packaging production, considering oscillations in the printing process and other relevant factors. Color differences $\Delta E^*$ of less than 2 cannot be seen with the naked eye, and differences of 4 may be acceptable to most people. For certain processes, $\Delta E$ differences of 4 to 8 are considered permissible.

The research results lead to the conclusion that ink I1, which contains about 80% renewable raw materials, achieves the best colorimetric properties in the specified experimental conditions, which can contribute to environmental sustainability in this area. The application of the results is significant in the field of sustainable circular eco-design when creating a graphic product; it contributes to new formulations of graphic materials with greater ecological suitability and is useful for closing the circular flow of materials; therefore, it is significant in the field of ecological sustainability of offset printing when it comes to cardboard packaging products.

**Author Contributions:** Conceptualization, I.B.M. and Z.B.; methodology, I.B.M., Z.B. and M.R.; validation, G.M. and M.R.; formal analysis, G.M. and M.R.; investigation, I.B.M. and Z.B.; data curation, G.M. and M.R.; writing—original draft preparation, I.B.M. and Z.B.; writing—review and editing, I.B.M. and Z.B.; visualization, I.B.M. and G.M.; supervision: Z.B. All authors have read and agreed to the published version of the manuscript.

**Funding:** This research received no external funding.

**Institutional Review Board Statement:** Not applicable.

**Informed Consent Statement:** Not applicable.

**Data Availability Statement:** The data that support the findings of this study are available from the authors upon reasonable request.

**Conflicts of Interest:** The authors declare no conflicts of interest.

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
