# Peer review of "Environmentally Sustainable Offset Prints Exposed to Thermal Aging and NO2"

_sustainability, doi:10.3390/su16041681_

Round 1

Reviewer 1 Report

Comments and Suggestions for Authors

Dear Authors,

Thank you for your submission of the manuscript. The paper addresses a crucial topic concerning the impact of environmentally sustainable offset prints exposed to thermal aging and NO2, rendering it particularly intriguing. Considering this review, I offer the following suggestions:

1. Please incorporate the study's objective as the initial sentence in the abstract to augment the rationale for the publication worthiness of this research.

2. Please extend the Introduction section by providing additional relevant information about the study.

3. Please add more references to the Introduction section by including them within lines 66 to 95.

4. Please try to improve the clarity of the presentation by dividing the Results and Discussion section into two distinct segments.

5. The authors are encouraged to quantify the measurement uncertainty by employing methods such as confidence intervals.

6. Please expand the list of references within lines 425 to 470.

7. Please add more relevant references in the Discussion and elaborate on the Discussion section comparing your study and other relevant research.

8. Please include a section highlighting the limitations of the study.

9. In the Conclusion section, articulate the significance of the study. For conciseness, consider utilizing bullet points to enhance clarity.

Thank you for your attention to these matters, and I wish you success in refining your work.

Best regards

Author Response

Dear reviewer,

We are grateful for the reviewer's words of praise. As for the reviewer's comments, the answers are listed in the order of reference:

  1. Include the aim of the study as an opening sentence in the abstract to further explain the rationale for publishing this research.

Author's answer:

At the beginning of the summary of the paper, we put a sentence that describes the key topic of the work and the justification for its publication.

  1. Please expand the Introduction section by providing additional relevant information about the study.

Author's answer:

We have expanded the Introduction by providing additional information as follows

  • Permanence primarily refers to the chemical resistance of the impression component and the influence of external factors. Durability depends on the characteristics of raw materials and the material production process as well as environmental conditions. These studies include the influence of the type, proportion and characteristics of cellulose fibres, fillers, sizing agents and additives for cardboard as a printing substrate. For the inks used, the research primarily refers to the type and proportion of solvents and the characterization of the yellow pigment. All the above is in scald with the requirements of offset printing, as a technique used to prepare prints, which are used as samples.
  • Research in the field of colourimetry is based on the CIE LAB colour space. This space is three-dimensional and is defined as a rectangular coordinate system with the edges L* (indicates brightness), a* (red-green) and b* (yellow-blue).
  • The paper presents the results of research related to colour attributes, but some talk about the experience of colour itself. On the print, the tone value is a term expressed in percentages used to indicate the visual experience of the shade concerning the printing surface and the value of the solid colour. When the base value is zero, it means that there is no tone, and the solid tone value is 100% Raster Tone Value (RTV), i.e. full tone. The tone of ink or tonality of colour of ink quality is determined by the wavelength of light rays which cause the sensation in the eye and colour. The colour of tone indicates the type of ink or ink itself.

  1. Add more references to the Introduction section by including them on lines 66 through 95.

Author's answer:

References have been added as recommended.

  1. Try to improve the clarity of the presentation by dividing the Results and Discussion section into two distinct segments.

Author's answer:

The work is divided into sections:

Results: 218-418

Discussion of results: 418-497

  1. The authors are encouraged to quantify the measurement uncertainty by employing methods such as confidence intervals.

Author's answer:

Tables 2, 3, and 4 have been added showing the variance and standard deviation for samples SI1, SI2, and SI3, 100% RTV for accelerated thermal aging and accelerated thermal aging with 100 ppm NO2. and 800 ppmNO2, and the measured values of the chromatic coefficients a*, b* and L* are included based on 10 measurements for each measurement data. Not all results have been shown due to the comprehensiveness of the research.

  1. Please expand the list of references within lines 425 to 470.

Author's answer:

According to the reviewer's suggestion, we have increased the number of references.

  1. Please add more relevant references in the Discussion and elaborate on the Discussion section comparing your study and other relevant research.

Author's answer:

The discussion of the results has been elaborated more appropriately and literature citations have been added.

  1. Please include a section highlighting the limitations of the study.

Author's answer:

The limitation of the study refers to the type of graphic product. For example, if packaging for food products or the pharmaceutical industry is produced, there are special conditions for selecting graphic materials about safety conditions. 497-499

  1. In the Conclusion section, articulate the significance of the study. For conciseness, consider utilizing bullet points to enhance clarity.

Author's answer:

The application of the results is significant in the field of sustainable circular eco-design when creating a graphic product, it contributes to new formulations of graphic materials with greater ecological suitability, it is useful for closing the circular flow of materials, so it is significant in the field of ecological sustainability of offset printing when it comes to cardboard packaging products.  550-554

Best regards

Autors of paper

Reviewer 2 Report

Comments and Suggestions for Authors

The manuscript authored by Ivana Bolanča Mirković et. al. is addressing a very important and timely subject about sustainable printing practices within the framework of a circular economy. The focus on how thermal ageing and NO2 exposure influence the colorimetric properties of environmentally friendly cardboard and offset inks is particularly noteworthy. While the study of environmental effects on print materials is not a completely new field, the unique combination of factors studied here brings a fresh perspective. However, it would be more beneficial if the research could delineate more clearly how it diverges from the existing studies in this field. I would suggest some minor modifications before publication.

Additional comments are as follows:

  1. 1. The research provides empirical data and insights into the sustainability and durability of print materials in the face of environmental challenges. Nevertheless, the manuscript could articulate its contribution more explicitly by discussing how these findings fill gaps in the current literature or pave new paths for future research.

  2. 2. Some sections contain complex sentences and technical jargon that might obscure clarity, especially for readers not intimately familiar with the subject. Simplifying these parts and providing clear definitions of key terms at their first introduction could significantly enhance the readability and clarity. Here are a couple of examples: A. "The resulting relationship is SI1<RI3<SI2." This statement, while concise, presents a relationship without adequate context or explanation. It assumes a level of familiarity with what SI1, RI3, and SI2 refer to, and why their relationship is significant. Expanding on this with more context would greatly enhance clarity and accessibility. B. "The research includes prints in full tone, 70% RTV, 50% RTV, and 30% RTV, and the intensity of the tonal experience will depend on the interaction of the substrate with the raster and different types of ink with raster print as a function of the experimental conditions." Terms like "full tone," "RTV," "tonal experience," "interaction of the substrate with the raster," and "different types of ink with raster print" are quite technical. Providing brief explanations or simplifying these terms could make the content more accessible.

  3. 3. The methodology of the study appears robust, and the argument is logically constructed upon the data collected and analyzed. The transition from introducing the problem and its significance to presenting and discussing the results demonstrates a coherent structure. However, the manuscript would benefit from a more detailed discussion of how the findings correlate with or challenge existing theories and knowledge. Please consider adding corresponded components.

Comments on the Quality of English Language

The manuscript's language is generally clear and conveys the complex technical content, please address the complex sentences and technical jargon problems listed in previous section.

Author Response

Dear reviewer,

We are grateful for the reviewer's words of praise. As for the reviewer's comments, the answers are listed in the order of reference:

  1. The presentation of the data should be improved. The stacked bar charts are difficult to read, and the reader should be able to follow each sample over time. A plot of different time series might be more effective.

Author's answer:

All diagrams have been transferred to simple bar Figures 2-13.

  1. The authors present no theory in their results. The paper is purely a measurement exercise. It would be interesting to understand, for example, why sample 2 deteriorates faster than samples 1 and 3 in the beginning, but then sample 3 deteriorates more subsequently. Is this something inherent about the properties of the materials that the inking prints are made of? Or is it a pure coincidence that some inks are better than others? In the first case – this should give us some indication as to which materials are more promising for future sustainable printing applications. In the second case, the prescription would be to test all available printing inks and find which one has the best properties. I want to stress that this comment is very important, as this distinguishes science from pure trial-and-error.

Author's answer:

We have divided the paper into the results section and the Analysis section. In the discussion, the essential settings concerning the composition of yellow offset inks were presented. Discussion lines 427-490. Why some colours fade faster probably depends on the kinetics of the chemical reaction, and our further research is headed in that direction.

  1. The authors should quantify the uncertainty of their measurements, using, for example, confidence intervals.

Author's answer:

The paper shows the standard deviation and variance for the selected samples, because we have abundant material that cannot be shown all. Tables 2-4

  1. If the authors make their paper about sustainability, I expect a sort of cost-benefit calculation at the end. More durable printing inks might be more resource-intensive, use more toxic or environmentally unfriendly inks, or have other environmental impacts regarding their recyclability. These must be weighed against the better durability achieved by some of the samples in this research

Author's answer:

It presented only one part of our extensive research in this area with the topic of ecological sustainability. We have published a paper in the area of recyclability and closing the circular flow. Your proposal is interesting, we have some knowledge about it, but we have also reduced this segment of the results presentation to satisfy the volume of one article.

Best regards

Authors of paper

Reviewer 3 Report

Comments and Suggestions for Authors

This paper presents and discusses how thermal ageing along with pollutant (NO2) exposure affects the stability and colorimetric characteristics of different printing inks.

Several comments are listed, in no particular order:

1. The presentation of the data should be improved. The stacked bar charts are difficult to read, and the reader should be able to follow each sample over time. A plot of different time series might be more effective.

2. The authors present no theory in their results. The paper is purely a measurement exercise. It would be interesting to understand, for example, why sample 2 deteriorates faster than samples 1 and 3 in the beginning, but then sample 3 deteriorates more subsequently. Is this something inherent about the properties of the materials that the inking prints are made of? Or is it a pure coincidence that some inks are better than others? In the first case – this should give us some indication as to which materials are more promising for future sustainable printing applications. In the second case, the prescription would be to test all available printing inks and find which one has the best properties. I want to stress that this comment is very important, as this distinguishes science from pure trial-and-error.

3. The authors should quantify the uncertainty of their measurements, using, for example, confidence intervals.

4. If the authors make their paper about sustainability, I expect a sort of cost-benefit calculation at the end. More durable printing inks might be more resource-intensive, use more toxic or environmentally unfriendly inks, or have other environmental impacts regarding their recyclability. These must be weighed against the better durability achieved by some of the samples in this research.

Overall, the opinion of the reviewer is that this paper is not a great fit for sustainability. Its motivation and connection to sustainability hinges on a brief exposition of the concept of a circular economy. The paper claims that it contributes to the goal of achieving a circular economy because it improves the durability of a product (cardboard prints) exposed to pollution. I do not see the connection between more durable printing inks and sustainability or a circular economy for several reasons: Firstly, there is no market failure to address here – consumers, firms, and environmentalists should all be aligned on the goal that more durable and higher quality inks are better. Secondly, the research is quite narrow in its scope: It researches the durability of a single product, through three different readily available technologies (i.e., inks). The insights of this research are thus only relevant to printing firms using these types of inks, and that haven’t already done their own internal R&D on the durability of their inks.

Comments on the Quality of English Language

Can be improved slightly. 

Author Response

(The authors gave the same response as above.)

Round 2

Reviewer 1 Report

Comments and Suggestions for Authors

Dear Authors,

I appreciate the modifications you made to the manuscript. With these revisions, I believe the paper has reached a level that makes it suitable for publication. Your efforts have significantly enhanced the overall quality, and I'm confident that the content is now well-prepared for the next steps in the publication process.

Best regards